# OpenReview forum: "Safeguarding System Prompts: A Surrogate-Based Defense Against Injection Attacks"
_ICLR.cc/2025/Conference — ICLR 2025 Conference Withdrawn Submission_

### Official Review · Reviewer_pZTV · 2024-10-18

**Soundness:** 3
**Presentation:** 2
**Contribution:** 1
**Rating:** 3
**Confidence:** 5

**Summary:**

The author proposed a new defense mechanism namely 'SurF' to act as a raw-prompt agnostic detector against prompt-injection-based stealing attacks, which is exclusive for scenarios where the defender has no access to the original system prompts as they may be confidential. 'SurF' exploits a bunch of surrogate prompts to test the user query and identifies the malicious query according to the output of the LLM. To evaluate the performance of the defense, the author proposed RFM and ACC as the metrics to show functional preservation and robustness. Extensive experiments demonstrated the effectiveness of the proposed method across a variety of mainstream LLM in comparison with the baseline where the defender is able to reach the system prompt.

**Strengths:**

1. Good writing, easy to follow.

2. The proposed method is simple yet effective, making it easy to reproduce.

3. SurF is a training-free method, it can be implemented easily.

**Weaknesses:**

1. This paper is based on an impractical threat model by assuming the defender is ignorant of the system prompt. In a real-world implementation, the system prompts are usually processed by the user or the service provider according to my knowledge. In the former case, there is no need for the prompt stealing attack as the user himself is aware of the prompt. For the latter, there is no need to use 'SurF' as the system prompt is accessible. Moreover, in either case, the service provider *MUST* have access to the system prompt otherwise it cannot be fed into the LLM. In case I misunderstood the scenario where the proposed method is supposed to function, I strongly suggest the author give a detailed real-world example illustrating why the system prompt can be inaccessible to both the service provider and the user.

2. I believe that the detecting mechanism of 'SurF' can be easily defeated when the stealing prompt asks the LLM to give trivially encrypted outputs, e.g. in Caesar cipher, which can result in a low CS score as well as a low WR score, so as to break 'SurF'. Note that the LLMs are capable of yielding such outputs according to [1,2].

3. The number of surrogate prompts K may influence the performance of SurF from Table 2, seemingly increasing the query cost by 5~10 times.

4. Only simple attacks and baselines are evaluated in the paper, an effective defense is supposed to be robust to the SOTA attacks [3] as well as the adaptive attacks (as narrated above).


[1] Glukhov, D., Shumailov, I., Gal, Y., Papernot, N., & Papyan, V. Position: Fundamental Limitations of LLM Censorship Necessitate New Approaches. In Forty-first International Conference on Machine Learning.

[2]  Yuan, Y., Jiao, W., Wang, W., Huang, J.-t., He, P., Shi, S., and Tu, Z. Gpt-4 is too smart to be safe: Stealthy chat with llms via cipher. In The Twelfth International Conference on Learning Representations, 2023

[3] Sha, Zeyang, and Yang Zhang. "Prompt stealing attacks against large language models." arXiv preprint arXiv:2402.12959 (2024).

**Questions:**

1. In which scenario is the proposed threat model practical? Does the author indicate one in which both the service provider and the user have no access to the system prompt? If so, why is it realistic?

2. Will SurF remain robust if the outputs are in cipher? If not, how to improve the proposed method?

3. Will SurF significantly increase the inference cost? How much is the extra query cost supposed to increase?

4. How robust is SurF against the SOTA attacks? And how resilient is it towards the mentioned adaptive attack?

5. The y-axis and the values in Figure 3 do not seem to be correct. The first row of the heatmap seems to refer to the performance when the LLM behind the service is Vicuna-7b, according to the APP values and the relative improvements as they sum up to 24.67, which is the no defense APP of Vicuna-7b in Table 1. I am confused to see that, in this case, the APP becomes even higher when exploiting some different proxy model (e.g., in the first row, second column it gets 27.29). What is your explanation for this phenomenon?

---

> ### Author Response · Authors · 2024-11-24
> **Official Comment by Authors**
>
> We sincerely thank the reviewer pZTV for taking the time to review our work. We are grateful for your recognition of our method as not only straightforward and effective but also training-free, which facilitates ease of implementation. In response to your insightful comments, we have provided detailed feedback below.
>
> ***Q1:*** Practicality of the Proposed Threat Model
> > This paper is based on an impractical threat model by assuming the defender is ignorant of the system prompt. In a real-world implementation, the system prompts are usually processed by the user or the service provider according to my knowledge. In the former case, there is no need for the prompt stealing attack as the user himself is aware of the prompt. For the latter, there is no need to use 'SurF' as the system prompt is accessible. Moreover, in either case, the service provider MUST have access to the system prompt otherwise it cannot be fed into the LLM. In case I misunderstood the scenario where the proposed method is supposed to function, I strongly suggest the author give a detailed real-world example illustrating why the system prompt can be inaccessible to both the service provider and the user. In which scenario is the proposed threat model practical? Does the author indicate one in which both the service provider and the user have no access to the system prompt? If so, why is it realistic?
>
> ***Ans for Q1:*** We understand your concern regarding the practicality of a threat model where the defender does not have access to the system prompt. Our proposed setting addresses scenarios involving third-party defenses where the system prompt is maintained confidentially by the user or service provider.
> For example, consider a scenario where a company owns a powerful Large Language Model (LLM) and a user possesses proprietary, effective, and secret system prompts essential for their business operations. The user wishes to integrate the LLM with their private prompts to create a customized business API. To protect the confidentiality of these system prompts, the user may engage a third-party security provider to implement SurF as a defense mechanism. In this case, the third-party defender does not have access to the internal LLM or the confidential system prompts, ensuring the prompts remain secure while still providing an effective defense against prompt-injection attacks.
> This setting is practical in commercial environments where protecting intellectual property and proprietary system prompts is crucial. The separation of LLM ownership and prompt ownership necessitates a defense mechanism that can operate without direct access to sensitive components, which SurF is designed to fulfill.
>
> ***Q2:*** Defense Against More Prompt Injection Attacks
> >  I believe that the detecting mechanism of 'SurF' can be easily defeated when the stealing prompt asks the LLM to give trivially encrypted outputs, e.g. in Caesar cipher, which can result in a low CS score as well as a low WR score, so as to break 'SurF'. Note that the LLMs are capable of yielding such outputs. Will SurF remain robust if the outputs are in cipher? If not, how to improve the proposed method?
>
> ***Ans for Q2:*** We acknowledge the potential for attackers to employ encryption techniques, such as Caesar cipher, to obfuscate the system prompt in their responses, potentially evading SurF’s detection mechanisms. In our current implementation, SurF primarily targets attacks where the system prompt is directly or semantically embedded in the LLM’s output.
> However, our approach includes calculating embedding similarity between surrogate prompts and responses (as per Equation (5)), which provides a measure beyond simple string matching. This semantic similarity helps in detecting paraphrased or translated versions of the system prompt. While this does enhance SurF’s ability to identify obfuscated prompts, we recognize that more sophisticated encryption-based attacks may require additional defenses. Future work will explore integrating more advanced detection mechanisms, such as pattern recognition or machine learning-based anomaly detection, to further enhance SurF’s resilience against such adaptive attacks.

---

> ### Author Response · Authors · 2024-11-24
> **Official Comment by Authors**
>
> ***Q3:*** Inference-Time Cost
> > The number of surrogate prompts K may influence the performance of SurF from Table 2, seemingly increasing the query cost by 5~10 times. Will SurF significantly increase the inference cost? How much is the extra query cost supposed to increase?
>
> ***Ans for Q3:*** We appreciate your concern regarding the inference-time cost associated with using multiple surrogate prompts. To mitigate this, SurF offers flexibility in choosing surrogate models. Instead of relying solely on large, computationally intensive LLMs, SurF can utilize smaller, more affordable surrogate models to perform the necessary checks. This approach significantly reduces the additional query cost while maintaining effective defense capabilities.
> Furthermore, the trade-off between computational overhead and security is a critical consideration. Protecting the confidentiality of system prompts is paramount, and in many commercial applications, a slight increase in inference time is acceptable compared to the potential commercial losses resulting from prompt leakage. Optimizing the number of surrogate prompts (K) and selecting efficient surrogate models are areas we continue to explore to enhance SurF’s practical applicability.
>
> ***Q4:*** Robustness Against SOTA and Adaptive Attacks
> > Only simple attacks and baselines are evaluated in the paper, an effective defense is supposed to be robust to the SOTA attacks as well as the adaptive attacks (as narrated above). How robust is SurF against the SOTA attacks? And how resilient is it towards the mentioned adaptive attack?
>
> ***Ans for Q4:*** We appreciate your insightful comment regarding the evaluation of our defense against adaptive adversaries who may attempt to evade detection by altering the prompt's appearance. In our current study, we have focused on manually crafted attacks, including direct, translated, and interspersing-based attacks, as detailed in Section 4.5. Additionally, our SurF method incorporates semantic similarity measures (as per Equation (5)), which help mitigate the risk of such transformed prompts bypassing the filtering system.
> While our approach is effective against these types of attacks, we acknowledge that it may not fully protect against more sophisticated, learning-based attacks that reconstruct prompts without direct disclosure. Evaluating the robustness of SurF against adaptive adversaries remains an important area for future research, and we are keen to explore these scenarios to further enhance the effectiveness of our defense mechanism. Nevertheless, our primary contribution lies in the innovative use of surrogate system prompts to detect and filter malicious inputs without requiring direct access to the original system prompts. We believe our work can help open avenues for future research to enhance and build upon this foundational approach.
>
> ***Q5:*** Clarification of Figure 3
> > The y-axis and the values in Figure 3 do not seem to be correct. The first row of the heatmap seems to refer to the performance when the LLM behind the service is Vicuna-7b, according to the APP values and the relative improvements as they sum up to 24.67, which is the no defense APP of Vicuna-7b in Table 1. I am confused to see that, in this case, the APP becomes even higher when exploiting some different proxy model (e.g., in the first row, second column it gets 27.29). What is your explanation for this phenomenon?
>
> ***Ans for Q5:*** Figure 3 illustrates the Attack Success Rate (APP) for SurF when utilizing different surrogate models across various LLM services. The X-axis represents the surrogate models employed by SurF, while the Y-axis denotes the LLMs behind the services. Each block in the heatmap indicates the APP when a specific surrogate model (column) is used to detect attacks against a particular LLM service (row). For example, the first value of 24.33 under the Vicuna-7b surrogate model and GPT-4 service indicates that SurF reduces the APP from 24.69 (as shown in Table 1, without defense) to 24.33.
> It is also interesting to observe that using different surrogate models can influence the effectiveness of the defense mechanism. Additionally, a reduction in APP does not necessarily imply an overall performance gain of the defense system, as there may be instances of false alarms or stricter thresholds leading to more inputs being classified as malicious. Nevertheless, this experiment demonstrates SurF’s flexibility. The ability to utilize various surrogate models allows the defense mechanism to operate effectively even when the exact LLM behind a service is unknown or computationally intensive.

---

### Official Review · Reviewer_HU3p · 2024-10-27

**Soundness:** 1
**Presentation:** 4
**Contribution:** 1
**Rating:** 3
**Confidence:** 5

**Summary:**

This paper propose a defense method towards an injection attack in which attackers use prompt to trigger the adversary to real the system prompt. In order to mitigate the attack, the authors adopt an idea to detect whether the response of the LLM contains system prompt. In the scenario that system prompt is not available, the authors propose to use surrogate system prompts for detection. The authors propose two ways: i) use string matching to see whether the surrogate system prompts match the LLM output. ii) calculate embedding similarity to see whether the surrogate prompts match the LLM output.

**Strengths:**

1. The idea is very easy to understand, which can easily be accessible to general audience.

2. The figures are aesthetic.

**Weaknesses:**

1. The attack motivation is not clear. Why the attackers want to extract the system prompt?

2. The setting that the defender has no access to system prompt is not realistic. If defenders (like OpenAI) don't have access to system prompt, how should they input the full prompts to their LLM in delopyment?

3. Technical contribution is very limited. The defense method is more suitable for a technical blog, but definitely not for a scientific conference like ICLR.

**Questions:**

NA

---

> ### Author Response · Authors · 2024-11-24
> **Official Comment by Authors**
>
> We sincerely thank Reviewer HU3p for taking the time to review our manuscript and providing valuable feedback. We appreciate your recognition of the clarity and aesthetic quality of our presentation. Below, we address each of your concerns in detail.
>
> ***Q1:*** Clarification of Attack Motivation
> > The attack motivation is not clear. Why the attackers want to extract the system prompt?
>
> ***Ans for Q1:*** We understand the need to clearly articulate the motivation behind attackers attempting to extract system prompts. The system prompt plays a crucial role in guiding the behavior and responses of a Large Language Model (LLM). For service providers, the system prompt encapsulates proprietary strategies, guidelines, and operational frameworks that ensure the LLM performs effectively and aligns with specific business objectives. If an attacker successfully extracts this system prompt, they can replicate or manipulate the service, leading to significant commercial losses and potential misuse of the technology. Such an extraction undermines the competitive advantage and intellectual property of the service provider, making it a highly valuable target for adversaries.
>
> ***Q2:*** Realism of the Defender’s Lack of Access to System Prompts
> > The setting that the defender has no access to system prompt is not realistic. If defenders (like OpenAI) don't have access to system prompt, how should they input the full prompts to their LLM in deployment?
>
> ***Ans for Q2:*** We appreciate your concern regarding the realism of a defense mechanism operating without access to the system prompt. This scenario is particularly relevant in third-party defense settings. For instance, consider a situation where a company possesses a robust LLM and wishes to deploy it through a third-party service provider to create a customized business API. The company retains ownership of its proprietary system prompts, which are essential for the API's performance and confidentiality. In this case, the third-party defense provider is responsible for protecting these system prompts without having direct access to them or the internal mechanisms of the LLM. This separation ensures that the defense system can safeguard the system prompts while maintaining the integrity and functionality of the deployed LLM service. Such a setting is practical in commercial environments where confidentiality and security are paramount, and third-party expertise is leveraged to enhance protection without compromising sensitive information.
>
> ***Q3:*** Technical Contribution
> > Technical contribution is very limited. The defense method is more suitable for a technical blog, but definitely not for a scientific conference like ICLR.
>
> ***Ans for Q3:*** While our proposed defense method may appear straightforward, it addresses a novel and critical security challenge in the domain of LLMs. The primary contribution lies in the innovative use of surrogate system prompts to detect and filter malicious inputs without requiring direct access to the original system prompts. This approach is not only practical but also foundational, as it establishes a baseline for defending against prompt injection attacks in black-box scenarios. Our evaluation framework rigorously tests the effectiveness of SurF across various LLM platforms, demonstrating its potential as a viable defense mechanism. Moreover, our work opens avenues for future research to enhance and build upon this foundational approach, making it a valuable contribution to the scientific community. We believe that addressing these security concerns is essential for the responsible deployment of LLMs, thereby justifying its suitability for a scientific conference like ICLR.

---

> ### Comment · Reviewer_HU3p · 2024-11-27
> **Thanks for the rebuttal**
>
> Thanks for the rebuttal. I still think its contribution does not reach the bar of ICLR. Also, I still think the considered scenario is not realistic. Particularly, how can the third-party service provider does not have access to the system prompt but still can provide LLM service? It does not make sense. Given this, I am going to maintain my score because the rebuttal does no address my concern.

---

### Official Review · Reviewer_ecK8 · 2024-10-28

**Soundness:** 2
**Presentation:** 3
**Contribution:** 2
**Rating:** 3
**Confidence:** 4

**Summary:**

This paper presents the following defense against prompt injection attacks that aim to extract the system prompt: run the input multiple times with surrogate prompts instead of the real system prompt and check if the surrogate prompts appear in the response.  If they do, reject the input.

**Strengths:**

- Simple, straightforward idea
- Defense can be run as a filter on top of a black-box system because it does not require knowledge of the actual system prompt

**Weaknesses:**

- The proposed defense assumes that the extracted prompt appears in the model's response.  This assumption is false for many prompt extraction methods: they reconstruct the prompt from responses to normal queries or model logits, etc.  See Morris et al. "Language Model Inversion" (ICLR 2024), Sha and Zhang "Prompt Stealing Attacks Against Large Language Models" (arXiv), Zhang et al. "Extracting Prompt by Inverting LLM Outputs" (EMNLP 2024), Shen et al. "Prompt Stealing Attacks Against Text-to-Image Generation Models" (USENIX Security 2024), He et al. "Automated Black-box Prompt Engineering for Personalized Text-to-Image Generation" (arXiv).

None of these papers are even cited in the submission, presenting a very partial and limited view of prompt extraction attacks.

- The proposed defense imposes a huge inference-time cost because every input needs to be re-run with multiple surrogate prompts.  This looks like a deal-killer for any practical deployment.

- There is no evaluation for adaptive adversaries who are aware of the defense.  For example, what if the adversary asks the system to return the system prompt but spell it backwards or in a different language, etc.  Since detection is based on looking for exact substrings of the prompt in the response, I don't think the defense works in this case.

**Questions:**

I think all of these are necessary before this submission can be accepted:

- Measurement of the inference-time cost of the defense (vs. other defenses and undefended systems).

- Evaluation of how effective the defense is against adversaries who are aware of the defense and try to evade it.

- Acknowledgment that the defense applies only to a narrow slice of prompt extraction attacks and is ineffective against attacks that reconstruct the system prompt rather than trick the system into disclosing it in the response.

---

> ### Author Response · Authors · 2024-11-24
> **Official Comment by Authors**
>
> We sincerely thank Reviewer ecK8 for dedicating time to review our manuscript and providing thoughtful and constructive feedback. We appreciate your recognition of the simplicity and practicality of our proposed SurF method, particularly its applicability as a filter on top of black-box systems without requiring knowledge of the actual system prompt. Below, we address each of your concerns in detail.
>
> ***Q1:*** Limited View of Prompt Extraction Attacks
> > The proposed defense assumes that the extracted prompt appears in the model's response. This assumption is false for many prompt extraction methods: they reconstruct the prompt from responses to normal queries or model logits, etc. See Morris et al. "Language Model Inversion" (ICLR 2024), Sha and Zhang "Prompt Stealing Attacks Against Large Language Models" (arXiv), Zhang et al. "Extracting Prompt by Inverting LLM Outputs" (EMNLP 2024), Shen et al. "Prompt Stealing Attacks Against Text-to-Image Generation Models" (USENIX Security 2024), He et al. "Automated Black-box Prompt Engineering for Personalized Text-to-Image Generation" (arXiv).
>
> ***Ans for Q1:***  We acknowledge your observation that our defense assumes extracted prompts appear directly in the model's responses, which may not hold true for more sophisticated prompt extraction methods such as those that reconstruct prompts from responses to normal queries or model logits. We appreciate you bringing relevant works to our attention. In our current setting, we consider attackers who can only modify inputs without access to the internal LLM or system prompts, focusing primarily on manually designed attacks such as direct attacks, translated attacks, and interspersing-based attacks. We agree that investigating our defense against more advanced, learning-based attacks is an important direction for future work, and we appreciate the suggestion to broaden our evaluation to include these methods.
>
> ***Q2:*** Inference-Time Cost of the Proposed Defense
> > The proposed defense imposes a huge inference-time cost because every input needs to be re-run with multiple surrogate prompts. This looks like a deal-killer for any practical deployment.
>
> ***Ans for Q2:*** Thank you for highlighting the concern regarding the inference-time cost associated with our proposed SurF method, which requires re-running inputs with multiple surrogate prompts. We recognize that this could pose practical challenges for deployment. However, the underlying LLM used in our SurF method is flexible; it can employ surrogate models to detect potential malicious attacks, as discussed in Section 4.3. This flexibility allows the defense mechanism to operate even when the exact LLM behind the service is unknown, making it feasible to use smaller and more affordable models to detect potential attacks.
> We believe that the additional inference-time cost represents a trade-off between computational efficiency and the enhanced privacy protection of system prompts. Given that the leakage of secret system prompts could result in significant commercial losses, we consider this trade-off to be a reasonable compromise for achieving a more secure system.
>
> ***Q3:*** Effectiveness Against Adaptive Adversaries
> > There is no evaluation for adaptive adversaries who are aware of the defense. For example, what if the adversary asks the system to return the system prompt but spell it backwards or in a different language, etc. Since detection is based on looking for exact substrings of the prompt in the response, I don't think the defense works in this case.
>
> ***Ans for Q3:*** We appreciate your insightful comment regarding the evaluation of our defense against adaptive adversaries who may attempt to evade detection by altering the prompt's appearance, such as by spelling it backwards or translating it into different languages. In our current study, we have focused on manually crafted attacks, including direct, translated, and interspersing-based attacks, as detailed in Section 4.5. Additionally, our SurF method incorporates semantic similarity measures (as per Equation (5)), which help mitigate the risk of such transformed prompts bypassing the filtering system.
> While our approach is effective against these types of attacks, we acknowledge that it may not fully protect against more sophisticated, learning-based attacks that reconstruct prompts without direct disclosure. Evaluating the robustness of SurF against adaptive adversaries remains an important area for future research, and we are keen to explore these scenarios to further enhance the effectiveness of our defense mechanism.

---

### Official Review · Reviewer_9Yq7 · 2024-11-04

**Soundness:** 2
**Presentation:** 2
**Contribution:** 2
**Rating:** 3
**Confidence:** 5

**Summary:**

The paper introduces SurF (Surrogate-based Filtering), an innovative method for defending against injection attacks on system prompts without direct access to them. SurF uses a surrogate prompt pool to identify potential leaks and classify harmful inputs, preventing responses from being generated. Experiments show SurF's effectiveness in reducing attack success across various LLMs. However, stronger defenses can impact the quality of legitimate responses, highlighting the need for balance between security and response consistency.

**Strengths:**

The paper introduces a comprehensive and practical approach, SurF, which simulates prompt-output interactions to detect malicious inputs without needing access to raw system prompts.
This makes it particularly well-suited for third-party defense scenarios where confidentiality is essential. Additionally, the extensive experiments conducted across multiple offline and online LLM platforms validate SurF's effectiveness in reducing prompt injection attack success rates.

**Weaknesses:**

1. There is significant ambiguity in the threat model. What capabilities should an attacker possess during the attack process? Can the attacker control the LLM? Is the attacker able to fine-tune the LLM?
2. The evaluation metric for data filtering is unclear. The paper employs the metric EXC (equation (1)) to identify exact matches of prompt leakage, marking any occurrence of system prompt sentences in the reply as a successful attack. However, for data filtering (equation (4)), the authors measure the percentage of system prompt sentences present in the reply and set an 80% threshold to detect attacks. This approach could introduce bias and unintentionally degrade the performance of data filtering. Why not use the same EXC metric for data filtering as well?
3. There is considerable ambiguity in the attack setting and design as described. According to the paper, the authors attempt to separate the LLM (target) from the defense mechanism, such that the defense system lacks access to both the LLM and the system prompts used within it. This raises questions about the specific situations or scenarios where the defense system would be unable to access the system prompts. Who would adopt and deploy such a defense? The paper does not provide a realistic scenario for this setup. In practical terms, it seems illogical to separate the defense from the LLM, as it is unusual for a defense to be managed by a third party who is neither familiar with the target LLM party nor trusted by it. Thus, the motivation for ensuring confidentiality in the proposed method makes less sense to me.
4. As a result, In the SruF framework, the authors propose a setting where the system prompt is inaccessible, which introduces significant ambiguity. It is unclear who is permitted to access the system prompt and why the defense mechanism is restricted from doing so. The paper fails to clarify who can access (or cannot access) what kind of information within the framework, leading to confusion. Moreover, if the system prompt is meant to be confidential and inaccessible to the defense party, it is contradictory that surrogate prompts are generated based on the system prompt. This creates substantial confusion and inconsistency within both the threat model and the system design.
5. Furthermore, the definition of confidentiality in the context of this paper is unclear. If the surrogate prompts are generated based on the original system prompt, they must inherently contain or convey information from that original prompt. This approach represents a compromise of confidentiality, at the point of cryptography.
6. Stronger attacks need to be considered. In the field of prompt injection attacks, certain studies, such as [1], use learning-based methods to initiate attacks. I recommend that the authors conduct experiments to test the performance of their defense mechanisms against these types of sophisticated attacks.

[1] Pasquini, Dario, Martin Strohmeier, and Carmela Troncoso. "Neural Exec: Learning (and Learning from) Execution Triggers for Prompt Injection Attacks." arXiv preprint arXiv:2403.03792 (2024).

**Questions:**

See above.

---

> ### Author Response · Authors · 2024-11-24
> **Official Comment by Authors**
>
> We would like to extend our sincere gratitude to Reviewer 9Yq7 for taking the time to thoroughly review our manuscript and provide insightful feedback. We appreciate your recognition of SurF’s innovative approach, its suitability for confidentiality-critical scenarios, and the comprehensiveness of our experiments. Below, we address each of your concerns in detail.
>
> ***Q1:*** Clarification of the Threat Model
> > There is significant ambiguity in the threat model. What capabilities should an attacker possess during the attack process? Can the attacker control the LLM? Is the attacker able to fine-tune the LLM?
>
> ***Ans for Q1:*** We apologize for any ambiguity regarding our threat model. In our setting, similar to the classic system prompt attacks discussed in [1], the attacker is restricted to interacting solely with the API. This means the attacker can only send raw inputs and receive outputs from the API without gaining access to any internal information about the API, including the type of the underlying LLM or any defense mechanisms in place. Additionally, the attacker does not have the capability to modify the LLM. This assumption ensures that the defense mechanism operates effectively within the defined security boundaries.
>
> ***Q2:*** Evaluation Metrics for Data Filtering
> > The evaluation metric for data filtering is unclear. The paper employs the metric EXC (equation (1)) to identify exact matches of prompt leakage, marking any occurrence of system prompt sentences in the reply as a successful attack. However, for data filtering (equation (4)), the authors measure the percentage of system prompt sentences present in the reply and set an 80% threshold to detect attacks. This approach could introduce bias and unintentionally degrade the performance of data filtering. Why not use the same EXC metric for data filtering as well?
>
> ***Ans for Q2:*** Thank you for pointing out the ambiguity in our evaluation metrics. The EXC metric is indeed consistent with previous works and measures exact matches of prompt leakage by identifying occurrences of system prompt sentences in the replies. For data filtering, we employ a word-level matching approach combined with our proposed SurF metric (Eq. 5), which assesses semantic similarity. This dual-metric approach enhances the defender’s ability to detect prompt leakage more effectively. The combination of exact matches and semantic similarity provides a more robust evaluation, reducing potential biases and improving the overall reliability of data filtering.
>
> ***Q3:*** Attack Setting and Practical Scenarios
> > There is considerable ambiguity in the attack setting and design as described. According to the paper, the authors attempt to separate the LLM (target) from the defense mechanism, such that the defense system lacks access to both the LLM and the system prompts used within it. This raises questions about the specific situations or scenarios where the defense system would be unable to access the system prompts. Who would adopt and deploy such a defense? The paper does not provide a realistic scenario for this setup. In practical terms, it seems illogical to separate the defense from the LLM, as it is unusual for a defense to be managed by a third party who is neither familiar with the target LLM party nor trusted by it. Thus, the motivation for ensuring confidentiality in the proposed method makes less sense to me.
>
> ***Ans for Q3:*** We appreciate your careful consideration of our attack setting and the practical scenarios where our defense mechanism would be applicable. In the third-party defense scenario, the defense system does not have access to the LLM or its internal system prompts. This setting is practical in commercial environments where, for example, a company possesses a powerful LLM, and a user has proprietary and effective system prompts that they wish to integrate with the LLM to create a customized business API. The success of this API relies on keeping the system prompts confidential. In such cases, the user may engage a third-party security company to design a defense system that protects these prompts without requiring access to the internal workings of the LLM. This approach ensures the confidentiality of the system prompts while maintaining the integrity and functionality of the API.

---

> > ### Author Response · Authors · 2024-11-24
> > **Official Comment by Authors**
> >
> > ***Q4:*** SurF Framework and System Prompt Confidentiality
> > > As a result, In the SruF framework, the authors propose a setting where the system prompt is inaccessible, which introduces significant ambiguity. It is unclear who is permitted to access the system prompt and why the defense mechanism is restricted from doing so. The paper fails to clarify who can access (or cannot access) what kind of information within the framework, leading to confusion. Moreover, if the system prompt is meant to be confidential and inaccessible to the defense party, it is contradictory that surrogate prompts are generated based on the system prompt. This creates substantial confusion and inconsistency within both the threat model and the system design.
> >
> > ***Ans for Q4:*** Thank you for pointing out the potential inconsistency regarding system prompt access within the SurF framework. To clarify, in our setting, system prompts are owned by individual creators who do not wish to grant access to the LLM provider or safety control entities. Consequently, surrogate prompts are generated independently by collecting various system prompts from open-source sources across different scenarios. This ensures that surrogate prompts do not contain or reveal information from the original, confidential system prompts. Exploring the generation of surrogate prompts based on secret system prompts, with the owner's consent, is indeed an interesting direction for enhancing defense capabilities without compromising confidentiality.
> >
> > ***Q5:*** Definition of Confidentiality
> > > Furthermore, the definition of confidentiality in the context of this paper is unclear. If the surrogate prompts are generated based on the original system prompt, they must inherently contain or convey information from that original prompt. This approach represents a compromise of confidentiality, at the point of cryptography.
> >
> > ***Ans for Q5:*** We acknowledge the confusion regarding the definition of confidentiality in our paper. In our framework, surrogate prompts are sourced from publicly available prompts and are distinct from the original, confidential system prompts. This separation ensures that surrogate prompts do not compromise the confidentiality of the original prompts. By generating surrogate prompts from diverse, non-confidential sources, we maintain the integrity and secrecy of the system prompts while still providing effective defense against injection attacks. This approach ensures that confidentiality is preserved, as surrogate prompts do not contain any sensitive information from the original system prompts.
> >
> > ***Q6:*** Consideration of Stronger Attack Methods
> > >  Stronger attacks need to be considered. In the field of prompt injection attacks, certain studies, use learning-based methods to initiate attacks. I recommend that the authors conduct experiments to test the performance of their defense mechanisms against these types of sophisticated attacks.
> >
> > ***Ans for Q6:*** We appreciate your suggestion to evaluate our defense mechanism against more sophisticated, learning-based attack methods. In our current setting, we have focused on manually designed attacks, including direct, translated, and interspersing-based attacks, under the assumption that attackers can only modify inputs without access to the internal LLM or system prompts. However, we recognize the importance of assessing our defense against learning-based attacks in black-box scenarios. Investigating the robustness of SurF against such advanced attack methods is indeed valuable, and we are keen to explore this in future research to further validate and enhance the effectiveness of our defense mechanism.
> >
> > > ***Reference***
> > >
> > > [1] Prompts should not be seen as secrets: Systematically measuring prompt extraction attack success. arxiv23

---

### Note · Authors · 2024-12-03

I have read and agree with the venue's withdrawal policy on behalf of myself and my co-authors.